# The Association between Proton Pump Inhibitors and the Effectiveness of CDK Inhibitors in HR+/HER- Advanced Breast Cancer Patients: A Systematic Review and Meta-Analysis

**DOI:** 10.3390/cancers15215133

**Published:** 2023-10-25

**Authors:** Yu-Cheng Chang, Junmin Song, Yu Chang, Chin-Hsuan Huang, Aarushi Sudan, Pei-Chin Chen, Kuan-Yu Chi

**Affiliations:** 1Department of Internal Medicine, Danbury Hospital, Danbury, CT 06810, USA; 2Department of Education, Center for Evidence-Based Medicine, Taipei Medical University Hospital, Taipei 110, Taiwan; 3Department of Medicine, Jacobi Medical Center, Albert Einstein College of Medicine, 1400 Pelham Parkway South, Building 1, 3N21, Bronx, NY 10461, USA; 4Section of Neurosurgery, Department of Surgery, National Cheng Kung University Hospital, College of Medicine, National Cheng Kung University, Tainan 704, Taiwan; 5Division of Gastroenterology and Hepatology, Department of Internal Medicine, Taipei Medical University Hospital, Taipei 110, Taiwan

**Keywords:** Palbociclib, Ribociclib, proton pump inhibitor (PPI), advanced breast cancer, survival rate

## Abstract

**Simple Summary:**

Proton pump inhibitors (PPIs) are frequently used in many patients with advanced breast cancer who are being treated with cyclin-dependent kinase inhibitors. In this study, our aim was to determine if there is an association between the use of PPIs and the effectiveness of cyclin-dependent kinase inhibitors, specifically Palbociclib and Ribociclib. We conducted a systematic review and meta-analysis through which to address this clinical question and found that the use of Palbociclib resulted in a lower survival rate for both overall and progression-free survival. However, the concurrent use of PPIs and Ribociclib did not lead to a decrease in the survival rates. Therefore, for cases in which advanced breast cancer patients require the use of PPIs, it may be reasonable to consider using Ribociclib.

**Abstract:**

There have been many clinical questions regarding whether the use of proton pump inhibitors (PPIs) could deteriorate the effects of cyclin-dependent kinase inhibitors (CDKIs) in HR+/HER2- advanced breast cancer patients. We performed a systematic review and meta-analysis of this clinical question, including studies enrolling HR+/HER2- metastatic breast cancer patients treated with CDKIs (Palbociclib or Ribociclib) and reporting at least one comparative survival outcome, either overall survival (OS) or progression-free survival (PFS), between concomitant PPI users and non-users. Eight studies met the eligibility criteria, with a total of 2584 patients included (PPI users: 830, PPI non-users: 1754), demonstrating that concomitant PPI use was associated with significantly higher risks of all-cause mortality (HR = 2.03; 95% CI, 1.49 to 2.77; I^2^ = 0%) and disease progression (HR = 1.75; 95% CI, 1.26 to 2.43; I^2^ = 59%) in breast cancer patients taking Palbociclib. In contrast, there were no significant survival impacts of PPIs on Ribociclib (HR = 1.46; 95% CI, 0.91 to 2.34; I^2^ = 36%). Additionally, there was no significant difference in the risk associated with CDKI dose reduction due to drug toxicity (RR = 1.12; 95% CI, 0.97 to 1.29). Therefore, when HR+/HER2- advanced breast cancer patients require the use of PPIs, it may be reasonable to consider using Ribociclib.

## 1. Introduction

Proton pump inhibitors (PPIs), which serve as gold-standard medications in the management of gastrointestinal symptoms in cancer patients [1] have been shown to compromise the effectiveness of various anti-cancer therapies, including immune checkpoint inhibitors and chemotherapy, due to their dysbiotic effects [2]. In addition to dysbiosis, the fluctuations in intragastric pH levels exacerbated by PPIs may alter the treatment response in those taking oral anti-cancer medications due to pH-dependent solubility, thereby leading to treatment failure [3]. Although a vast amount of evidence has illustrated that PPIs interfere with the oral bioavailability levels of various oral medications that are dependent on pH levels [4,5], it still remains uncertain whether or not such associations confer any meaningful clinical significance. In the case of epidermal growth factor (EGFR) inhibitors, for example, while several clinical trials have shown that long-term use of PPIs has little survival impact on patients treated with Erlotinib [6] and Osimertinib [7], the effectiveness of Pazopanib appears to be compromised by concurrent PPIs [8]. These conflicting results suggest that dysbiotic effects and intragastric pH interruption might not be the only factors when considering potential drug–drug interactions between PPIs and oral anti-cancer drugs; furthermore, other prognostic factors, such as cancer type, comorbidities, and gastrointestinal tract-related infections, should also be considered [9,10,11]. Moreover, more clinical studies pertaining to real-world data are needed to address the prognostic influences of PPIs on anti-cancer medications for different cancers.

Palbociclib and Ribociclib, which are cyclin-dependent kinase inhibitors (CDKIs), have been approved for the first-line treatment of hormone receptor-positive (HR+) and human epidermal growth factor receptor 2-negative (HER2−) metastatic breast cancer patients [12], as they have both demonstrated that they significantly prolong median progression-free survival (PFS) in advanced breast cancer patients, when combined with aromatase inhibitors [13]. Although Palbociclib and Ribociclib are both basic compounds, from a pharmacokinetic standpoint, Palbociclib is more likely to be affected by PPIs than Ribociclib because of their different solubilities [14,15,16]. However, clinical studies do not seem to support this perspective, as some of them found that PPIs had no prognostic influences on those taking Palbociclib [17,18,19], while others concluded that PPIs negatively affected the effectiveness of Ribociclib [20]. Given the increased use of Palbociclib and Ribociclib with no restrictions on the concomitant use of PPIs in breast cancer patients, and with incongruent results between the clinical studies mentioned above, we aim to conduct a systematic review and a meta-analysis through which to investigate the prognostic influences of PPIs on these two CDKIs in breast cancer patients. To the best of our knowledge, this study is the first-ever meta-analysis addressing this topic.

## 2. Materials and Methods

The systematic review and meta-analysis herein were conducted based on the Cochrane Handbook for Systematic Reviews of Interventions [21], and the subsequent outcomes were reported in accordance with the Preferred Reporting Items for Systematic Reviews and Meta-Analyses (PRISMA) (Appendix A). Additionally, this study has been registered in PROSPERO (CRD42023454552).

### 2.1. Study Selection

PubMed, Embase, the Cochrane Library, LARVOL Cancer Trial Results, and ClinicalTrials.gov were searched, from their inception dates to 31 July 2023. Three investigators (Y.C.C., J.S., and A.S.) independently identified relevant studies, with no restrictions placed on language or the country of publication. The investigators first performed title and abstract screenings, and discrepancies were addressed by reaching a consensus with a senior reviewer (P.C.C.). All plausibly eligible articles were then retrieved by the same investigators, and the full texts were reviewed. Any disagreement was discussed and resolved with another independent author (K.Y.C.).

Our search strategy included the following keywords: (“CDK inhibitor” OR “cyclin-dependent kinase inhibitors” OR “Palbociclib” OR “Ribociclib” OR “Abemaciclib”) AND (“proton pump inhibitors” OR “Omeprazole“ OR “Esomeprazole” OR “Lansoprazole” OR “Rabeprazole” OR “Pantoprazole” OR “Dexlansoprazole” OR “Zegerid” OR “gastric acid suppressant” OR “H2 blocker” OR “antihistamine” OR “H2 antagonist”). Regarding grey literature, the keywords (“hormone receptor positive breast cancer” AND “HER2 negative breast cancer”) AND (“palbociclib” OR “ribociclib”) were used for LARVOL Cancer Trial Results and ClinicalTrials.gov.

### 2.2. Eligibility Criteria

The predefined criteria for evidence selection were as follows: (1) randomized controlled trials (RCTs) with prospective or retrospective cohort studies; (2) studies involving adult patients aged over 18 with HER2-negative (score 0 or 1+ or negative with immunohistochemistry and negative staining with dual-probe in situ hybridization) and hormone receptor-positive (tumors with more than 1% estrogen receptor activity) metastatic breast cancer treated with a CDKI (Palbociclib or Ribociclib), with or without concomitant PPI use; (3) ‘concomitant PPI’ use was defined as any exposure to PPIs over the entire CDKI treatment duration; and (4) studies reporting at least one comparative survival outcome, either overall survival (OS) or progression-free survival (PFS), between PPI users and non-users, irrespective of indications.

### 2.3. Data Extraction

Two investigators (Y.C.C. and C.C.H.) independently extracted relevant information from eligible articles, including (1) the first author’s name and the publication year, (2) the study type, (3) the country, (4) the chemotherapeutic regimen, (5) the sample size, (6) the PPI administration window, (7) the PPI regimen with its corresponding dosage, (8) the ages of patients, (9) Eastern Cooperative Oncology Group performance status (ECOG-PS), and (10) both unadjusted and adjusted time-to-event survival outcomes, including OS and PFS, between PPI users and non-users (when studies did not report the hazard ratios (HRs), but presented Kaplan–Meier survival curves instead, we acquired an estimated HR from the curves through a well-established method [22], using the calculation spreadsheet developed by Tierney and colleagues [23]), and (11) CDKI dose reduction due to drug toxicity in both PPI users and non-users. 

### 2.4. Quality Assessment

Two investigators (J.S. and Y.C.) independently completed literature appraisal using the Cochrane Risk of Bias tool 2.0 [24] for RCTs, and the Risk of Bias in Non-randomized Studies of Interventions (ROBINS-I) [25] tool for non-RCTs. Any discrepancy was addressed through discussion with a third investigator (K.Y.C.).

For observational studies, it is imperative to note that, to mitigate immortal time bias and accurately investigate the association between concomitant PPI and CDKI use, PPIs, ideally, ought to be administered at the same time as CDKI is initiated (Figure 1A). Alternatively, patients should be followed up staring from the PPI index date, instead of the CDKI initiation date, if there is a potential immortal period (Figure 1B). Otherwise, any outcome event, such as disease progression or death, transpiring subsequent to the administration of CDKIs, but preceding the commencement of PPI administration, could be erroneously associated with the use of PPIs, thus rendering selection bias (Figure 1C). Therefore, our assessment of selection bias occurred in the following process: (1) if the date of cohort entry aligned with the date of PPI prescription, or if patients were followed up starting from the PPI index date, the risk was considered to be low; (2) if patients were followed up starting from the cohort entry date and there exists a substantial or unknown interval between the cohort entry date and the PPI index date, defined as exceeding half of the entire duration of CDKI treatment, the risk was deemed as high; (3) any scenario not meeting the aforementioned criteria was characterized as having a moderate risk of bias.

### 2.5. Main Outcomes and Statistical Analyses

Primary outcomes included OS and PFS between PPI users and non-users. Secondary outcomes constituted the risk ratios (RRs) of CDKI dose reduction due to drug toxicity between PPI users and non-users. The RRs obtained from each study were pooled using the Mantel–Haenszel method. All estimated effects were presented with a 95% confidence interval (CI). All meta-analyses were conducted using RStudio with the “meta” package (Appendix A). The pooled estimate was based on random effects with the restricted maximum likelihood (REML) [26] method, due to inevitable between-trial variance. Heterogeneity was assessed using I-square [27], with values of I^2^ < 25%, 25% < I^2^ < 50%, and I^2^ > 50% indicating low, moderate, and high heterogeneity, respectively. Pre-specified sensitivity analyses included subgroup analyses based on different CDKI regimens, using an adjusted HR, and the exclusion of studies subject to a serious risk of bias. We also performed leave-one-out analysis to investigate the robustness of pooled estimates and to examine whether pooled estimates were affected by any single study. The determination of statistical significance in these analyses followed the common threshold (*p* < 0.05). 

## 3. Results

After the systematic review, we identified a total of 384 references, including 18 from PubMed, 165 from Embase, 17 from MEDLINE, 7 from CENTRAL, 168 from LARVOL, and 9 from ClinicalTrials.gov. After duplicate exclusion, 148 studies were identified. Following the title screening and the removal of duplicates, 12 articles were selected for full-text inspection. References and studies related to the eligible articles were manually searched, and we added 4 more studies to the eligible articles. After full-text inspection, 8 studies were excluded. Ultimately, after the exclusion of ineligible articles, 8 studies fulfilled the eligibility criteria and were included for qualitative and quantitative syntheses (Figure 2).

### 3.1. Study Characteristics (Table 1)

We included a total of 2584 patients (PPI = 830; non-PPI = 1754) in our study. Seven [16,17,18,19,20,28,29] were retrospective studies and the remaining one, Cosimo 2023 [30], was a post hoc analysis of the PARSIFAL trial [31]. The window of PPI administration was consistently more than one-third of the entire CDKI treatment period across included studies, except for Çağlayan 2022 [17], in which the PPI treatment window was not defined. Most ECOG-PS scores of the patients enrolled in the included studies ranged from 0 to 2, with the exception of Odabas 2023 [18], which enrolled patients of ECOG 0–3.

**Table 1 cancers-15-05133-t001:** Study characteristics.

Included Studies	Nation	Study Type	Chemotherapeutic Regimen	Sample Size, *n*	First Line, *n* (%)	PPI, *n* (%)	PPI Use Window	Outcome Follow-Up	Age, Mean or Median (SD or IQR)	ECOG
	**Palbociclib**									
Cosimo 2023 [30]	Italy	Post hoc of RCT	Palbociclib + Fulvestrant/letrozole	416	416 (100)	91 (21.9)	PPI use started together with CDKI initiation; >2/3 of the treatment	From CDKI index date	63 (25–90)	0–2
Çağlayan 2023 [17]	Turkey	Retrospective	Palbociclib + Fulvestrant/Ai	36	0 (0)	16 (44.4)	N/A	From CDKI index date	PPI+: 55 (11.8) PPI−: 56 (13.9)	N/A
Lee 2023 [29]	Korea	Retrospective	Palbociclib + Fulvestrant/letrozole	1310	1310 (100)	344 (26.2)	>1/3 of the treatment	From PPI index date	N/A	N/A
Schieber 2023 [19]	USA	Retrospective	Palbociclib + Fulvestrant/letrozole/Tamoxifen	82	82 (100)	32 (39)	PPI use started together with CDKI initiation; >1/2 of the treatment	From CDKI index date	PPI+: 62.5 (53–68) PPI−: 68.5 (54–74)	0–2
Odabas 2023 [18]	Turkey	Retrospective	Palbociclib + Fulvestrant/letrozole	120	70 (58.3)	57 (47.5)	>2/3 of the treatment	From CDKI index date	PPI+: 60 (33–92) PPI−: 54 (25–86)	0–3
Eser 2022 [20]	Turkey	Retrospective	Palbociclib + Fulvestrant/letrozole	105	N/A	65 (61.9)	>1/2 of the treatment	From CDKI index date	PPI+: 61 (32–83) PPI−: 58 (36–76)	0–2
Del Re 2021 [16]	Italy	Retrospective	Palbociclib + Fulvestrant/letrozole	112	112 (100)	56 (50)	PPI use started before CDKI initiation; >2/3 of the treatment	From CDKI index date	PPI+: 63 PPI−: 61.5	0–2
	**Ribociclib**									
Çağlayan 2023 [17]	Turkey	Retrospective	Ribociclib + Fulvestrant/Ai	50	0 (0)	29 (58.0)	N/A	From CDKI index date	PPI+: 55 (11.8) PPI−: 56 (13.9)	N/A
Odabas 2023 [18]	Turkey	Retrospective	Ribociclib + Fulvestrant/letrozole	113	52 (52.0)	29 (25.7)	>2/3 of the treatment	From CDKI index date	PPI+: 60 (34–84) PPI−: 53 (31–80)	0–3
Del Re 2022 [28]	Italy	Retrospective	Ribociclib + Fulvestrant/letrozole	128	128 (100)	50 (39)	PPI use started before CDKI initiation; >2/3 of the treatment	From CDKI index date	PPI+: 64 PPI−: 58	0–2
Eser 2022 [20]	Turkey	Retrospective	Ribociclib + Fulvestrant/letrozole	112	N/A	61 (54.4)	>1/2 of the treatment	From CDKI index date	PPI+: 57 (38–87) PPI−: 49 (32–87)	0–2

PPI, proton pump inhibitor; SD, standard deviation; IQR, interquartile range; ECOG, Eastern Cooperative Oncology Group; CDKI, cyclin-dependent kinase inhibitor; Ai, aromatase inhibitor.

### 3.2. Risk of Bias Assessment (Appendix A)

Overall, six studies were assessed as having low [29,30] or moderate [16,18,19,28] risks of bias, while the other two [17,20] were evaluated as having serious risks of bias. 

We found that the proportion of ECOG-2 patients in Eser 2022 [20] was significantly higher in PPI users than in PPI non-users. As a consequence, the risk of bias in Eser 2022 was assessed as serious due to confounding.

Of note, there were unknown and significant intervals between the CDKI entry date and the PPI index date in Çağlayan 2022 [17] and Eser 2022 [20], respectively. Therefore, they were deemed to be subject to serious risk of selection bias. Although patients enrolled in Del Re 2022 [28] were followed up starting from the cohort entry date, the PPI use window covered at least two-thirds of the entire CDKI period. Therefore, the selection bias of Del Re 2022 [28] was graded as moderate risk.

### 3.3. Survival Association between PPIs and Palbociclib

Overall, three studies [18,29,30] and seven studies [16,17,18,19,20,29,30] reported the influences of PPIs on OS and PFS, respectively, in HR+/HER2- metastatic breast cancer patients taking Palbociclib. Our meta-analysis demonstrated that the concomitant use of PPIs was associated with a significantly higher risk of all-cause mortality (HR, 2.03; 95% CI, 1.49 to 2.77; I^2^ = 0%; Figure 3) and disease progression (HR, 1.75; 95% CI, 1.26 to 2.43; I^2^ = 59%; Figure 3) than the risk observed in non-users. Of note, our leave-one-out sensitivity analysis demonstrated that the pooled effect sizes were not affected by any single study (Appendix A). Additionally, sensitivity analyses using an adjusted HR (Appendix A), and excluding studies subject to serious risk of bias (Appendix A), conferred a similar association. 

To investigate the possible origin of heterogeneity in progression-free survival (PFS), we performed a subgroup analysis centered on the endocrine sensitivity profile. Nonetheless, this analysis revealed no notable distinction in disease progression magnitude between the endocrine-sensitive (HR, 1.79; 95% CI, 1.51 to 2.12; I^2^ = 7%; Figure 4) and endocrine-resistant cohorts (HR, 1.76; 95% CI, 1.15 to 2.68; I^2^ = 0%; Figure 4). 

### 3.4. Survival Association between PPIs and Ribociclib

No studies reported the OS between PPI users and non-users in patients taking Ribociclib. Conversely, a total of four studies [17,18,20,28] investigated the difference in PFS between PPI users and non-users in HR+/HER2- metastatic breast cancer patients receiving Ribociclib. Of note, there was no significant difference in disease progression between PPI users and non-users (HR, 1.46; 95% CI, 0.91 to 2.34; I^2^ = 36%; Figure 5). Notably, the leave-one-out sensitivity analysis showed consistently significant pooled effect sizes, suggesting that pooled estimates were not influenced by individual studies (Appendix A).

### 3.5. CDKI Dose Reduction

Our meta-analysis demonstrated that there was no significant difference in the risk of CDKI dose reduction due to drug toxicity (RR, 1.12; 95% CI, 0.97 to 1.29; I^2^ = 0%; Figure 6) between concomitant PPI users and non-users. Of note, the subgroup analysis based on the type of CDKI also showed no significant difference between Palbociclib (RR, 1.06; 95% CI, 0.89 to 1.27; I^2^ = 0%; Figure 6) and Ribociclib (RR, 1.24; 95% CI, 0.96 to 1.61; I^2^ = 0%; Figure 6). The leave-one-out sensitivity analysis indicated that pooled estimates were not influenced by individual studies (Appendix A).

## 4. Discussion

To the best of our knowledge, this is the first systematic review with meta-analysis that delves into the association between concurrent PPI and CDKI use in HR+/HER2- advanced breast cancer patients taking CDKIs. Our meta-analysis demonstrated that the use of concomitant PPIs was associated with significantly higher all-cause mortality and disease progression in in HR+/HER2- metastatic breast cancer patients taking Palbociclib as their CDKI, irrespective of their endocrine sensitivity profile. Our leave-one-out sensitivity analysis showed that our pooled estimates of desired outcomes were not affected significantly by individual studies. Moreover, other pre-specified sensitivity analyses also demonstrated similar results when using adjusted HR and excluding studies subject to serious risks of bias. All of these sensitivity analyses supported the robustness of the results of our meta-analysis. 

The mainstream mechanism accounting for the negative impact of concurrent use of PPIs on the effectiveness of Palbociclib is associated with pH-dependent solubility. It is conceivable that PPIs increase intragastric pH, reducing the absorption of weak base medications in a pH-dependent fashion [32]. Notably, the solubility of Palbociclib decreases to less than 0.5 mg/mL when intragastric pH rises above 4.5, which is within the capabilities of PPIs [16,28]. Resonating with this phenomenon, Sun et al. [14] revealed that, under fasting/fed states, the maximum concentration (Cmax) values and the areas under the concentration curve (AUCs) for Palbociclib were reduced by 62%/41% and 80%/13%, respectively, following the short-term use of rabeprazole. A previous study has also reported that the ratio of free average steady-state concentration (Css) to in vitro cell potency (Css/IC50) for Palbociclib was 0.94 [33]. As the Css for Palbociclib was similar to the IC50, even a slight alteration in drug absorption could reduce Palbociclib’s plasma level to below its minimum effective concentration, thus compromising clinical efficacy.

On the other hand, it is important to note that PPIs were shown to have little prognostic influence on those treated with Ribociclib. Although Ribociclib shares the same mechanism of action as Palbociclib and has a similar weak base property as well, it is not affected by concurrent PPIs, as Palbociclib is, which can be attributed to their distinctive chemical structures contributing to distinctive dissolution properties. As opposed to Palbociclib, a pyrido [2,3-d] pyrimidine analogue, Ribociclib is a 2-amino-pyr-rolo [2,3-d] pyrimidine derivative, which enables Ribociclib to maintain its solubility above 2.4 mg/mL when intragastric pH is over 4.5 [15,16]. In terms of pharmacokinetic parameters, Samant et at. reported no differences in Cmax and AUC after the ingestion of Ribociclib 600 mg with concurrent PPIs suggesting a negligible impact of gastric pH changes on Ribociclib bioavailability [15]. The fact that the Cmax and AUC values of Ribociclib remained unaffected by concurrent PPI use contrasted with the impacts of PPIs on the Cmax and AUC values of Palbociclib. This contrast provides a plausible explanation for the discrepancy in our observations between Palbociclib and Ribociclib. Furthermore, Ribociclib’s Css is far greater than the IC50, leading to a Css/IC50 ratio of more than 25 [15]. This wide therapeutic index minimizes the impact of drug absorption on clinical endpoints. Together with our meta-analysis, concomitant use of PPIs appears to be an independent covariate in breast cancer patients taking Ribociclib, in terms of both pre-clinical and clinical settings.

Although the aforementioned mechanism in pre-clinical studies seems to be plausible for supporting our clinical findings, one interesting phenomenon merits further discussion. Palbociclib and Ribociclib act as substrates for the efflux transporter P-glycoprotein (P-gp), and, as a matter of fact, PPIs are known as moderate inhibitors of P-gp [34]. In ex vivo studies, it was found that rabeprazole and omeprazole decreased the absorptive permeability of Palbociclib by 3.04- and 1.26-fold, respectively, and of Ribociclib by 1.76- and 2.54-fold, respectively, highlighting the ability of PPIs to inhibit the P-gp-mediated efflux of both CDKIs [35]. Although the PPI inhibition of CDKI efflux could theoretically increase the concentration of a CDKI, thereby increasing drug toxicity, our meta-analysis demonstrated that there was no difference in dose reduction due to drug toxicity between PPI users and non-users. It appears that the gastric pH elevation induced by PPIs may triumph over the inhibition of P-gp-mediated CDKI efflux, based on current pre-clinical and clinical evidence.

Our study possesses several strengths. Firstly, this is the first systematic review with meta-analysis investigating the interaction between PPIs and Palbociclib and Ribociclib in HR+/HER2- advanced breast cancer patients. Secondly, our meta-analysis shows that only Palbociclib users are significantly affected by concomitant PPI use, whereas PPIs have a minimal impact on Ribociclib users. Nevertheless, our study contains a myriad of limitations that ought to be addressed in the future. Firstly, most studies we included were observational studies, which may inherently introduce undetected confounders. Secondly, although this is the first systematic review with meta-analysis on this issue, the number of reviewed studies is relatively small (<10), which hinders us from assessing a potential small-study bias and publication bias. Thirdly, the ECOG-PS scores of the patients included in our study were mostly between 0 and 2, which jeopardizes the generalizability of our study results to other breast cancer patients. Fourthly, we were unable to investigate the individual influences of different PPIs on Palbociclib and Ribociclib, due to the limited data available from the included studies, which is vital information, as different PPIs vary in their levels of acid suppression. Fifthly, some outcomes had high heterogeneity, which could be ascribed to differences in study durations, patient characteristics, and control groups. Therefore, having a knowledge of their respective prognostic impacts can certainly help clinicians gain deeper insight when encountering such patients. Last but not least, indications for the use of PPIs are scarce, which prevents us from probing potential confounders in PPI users.

## 5. Conclusions

Our meta-analysis demonstrates that concomitant use of PPIs is associated with significantly higher rates of all-cause mortality and disease progression only in patients receiving Palbociclib. This negative association was not observed in patients taking Ribociclib. Also, the use of PPIs is not related to CDKI dose reduction. Therefore, when HR+/HER2- advanced breast cancer patients require the use of PPIs, it may be reasonable to use Ribociclib for CDKI treatment. Nonetheless, due to several limitations, discussed above, further high-quality prospective studies are warranted to elucidate the prognostic role of PPIs in CDKI use.

## Figures and Tables

**Figure 1 cancers-15-05133-f001:**
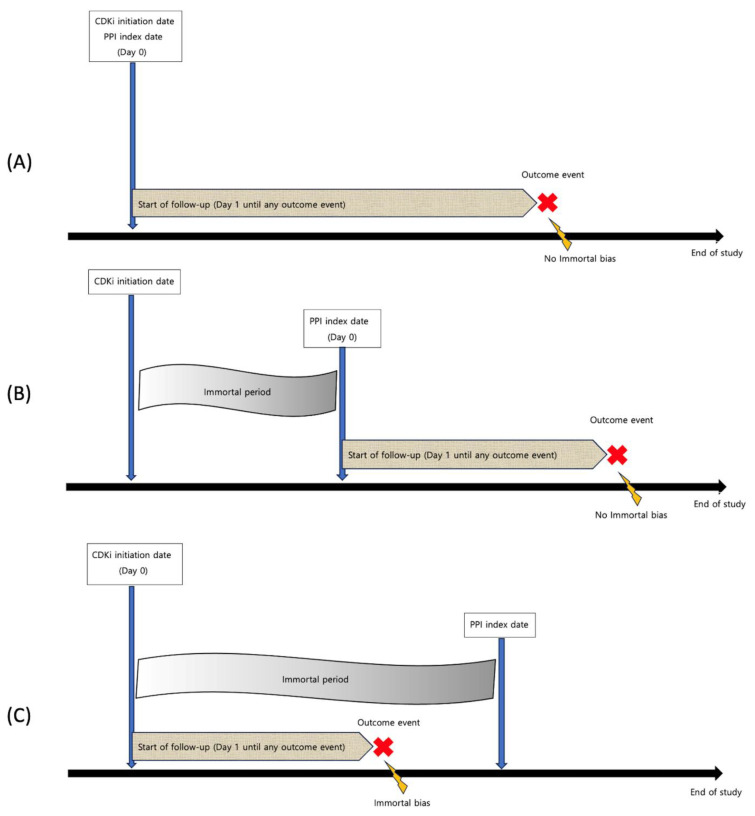
Potential immortal time bias. (**A**) There was no concern for immortal time bias if the PPI index date was the same as the CDKI initiation date. (**B**) If the PPI index date occurred after the CDKI initiation date, there was no concern for immortal time bias if patients were followed up from the time of starting PPIs. (**C**) If the PPI index date occurred after the CDKI initiation date, there was significant concern for immortal time bias if patients were followed up from the CDKI initiation date instead of the PPI index date.

**Figure 2 cancers-15-05133-f002:**
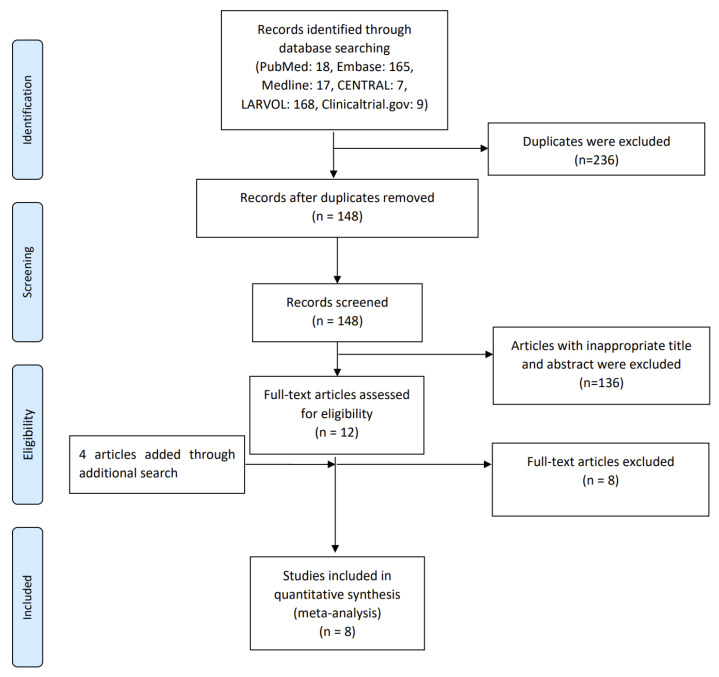
PRISMA flowchart diagram. We initially extracted a total of 384 potential references, including 18 from PubMed, 165 from Embase, 17 from MEDLINE, 7 from CENTRAL, 168 from LARVOL, and 9 from ClinicalTrials.gov. After duplicate exclusion, 148 studies were identified. Screening the titles and abstracts yielded 12 full-text articles which were assessed for their eligibility. After full-text inspection, 4 further studies were added, while 8 were excluded. Ultimately, 8 studies fulfilled the eligibility criteria and were included for qualitative and quantitative syntheses. PRISMA, Preferred Reporting Items for Systematic Reviews and Meta-Analyses.

**Figure 3 cancers-15-05133-f003:**
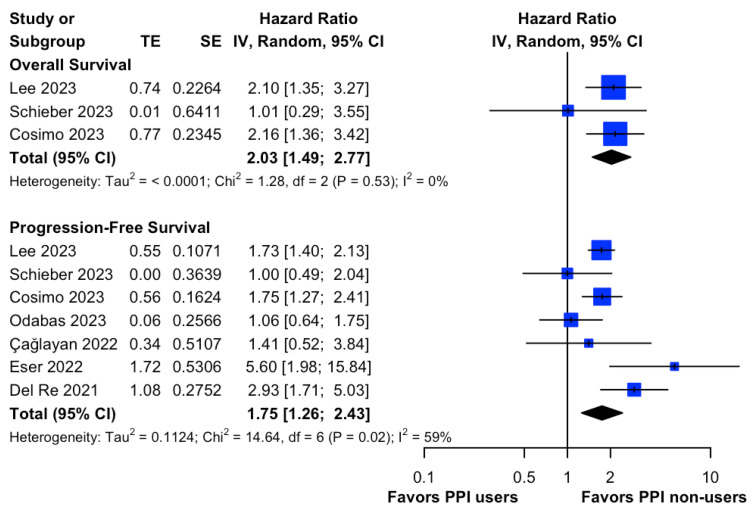
Forest plot of comparative overall survival and progression-free survival between PPI users and non-users in HR+/HER2- metastatic breast cancer patients taking Palbociclib. The size of each blue square is proportional to the weight of the indicated study. Horizontal lines indicate the 95% CI of each study; the black diamond indicates the pooled estimate of all studies after meta-analysis, with a 95% CI. TE is used to show the estimate of the treatment effect. PPI, proton pump inhibitor; CI, confidential interval; HR, hazard ratio; IV, inverse variance; SE, standard error. References: Lee (2023) [29], Schieber (2023) [19], Cosimo (2023) [30], Odabas (2023) [18], Çağlayan (2022) [17], Eser (2022) [20], Del Re (2021) [16].

**Figure 4 cancers-15-05133-f004:**
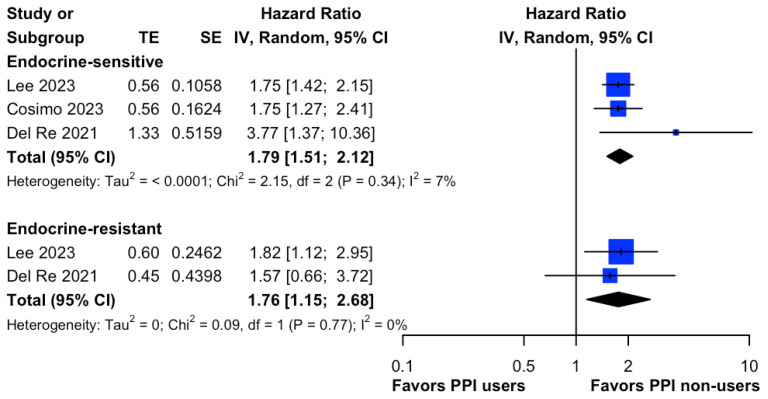
Forest plot of comparative progression-free survival between PPI users and non-users in endocrine-sensitive and endocrine-resistant HR+/HER2- metastatic breast cancer patients taking Palbociclib. The size of each blue square is proportional to the weight of the indicated study. Horizontal lines indicate the 95% CI of each study; the black diamond indicates the pooled estimate of all studies after meta-analysis with a 95% CI. TE is used to show the estimate of the treatment effect. PPI, proton pump inhibitor; CI, confidential interval; HR, hazard ratio; IV, inverse variance; SE, standard error. References: Lee (2023) [29], Cosimo (2023) [30], Del Re (2021) [16].

**Figure 5 cancers-15-05133-f005:**
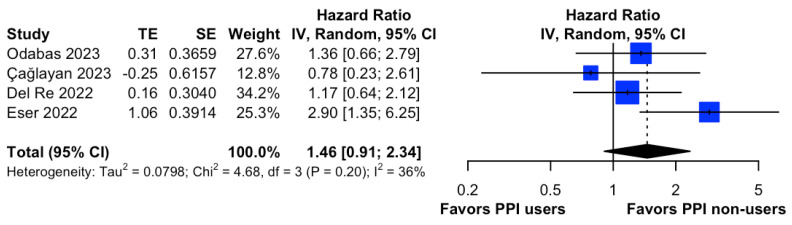
Forest plot of comparative progression-free survival between PPI users and non-users in HR+/HER2- metastatic breast cancer patients taking Ribociclib. The size of each blue square is proportional to the weight of the indicated study. Horizontal lines indicate the 95% CI of each study; the black diamond indicates the pooled estimate of all studies after meta-analysis with a 95% CI. TE is used to show the estimate of the treatment effect. PPI, proton pump inhibitor; CI, confidential interval; HR, hazard ratio; IV, inverse variance; SE, standard error. References: Odabas (2023) [18], Çağlayan (2022) [17], Del Re (2022) [28], Eser (2022) [20].

**Figure 6 cancers-15-05133-f006:**
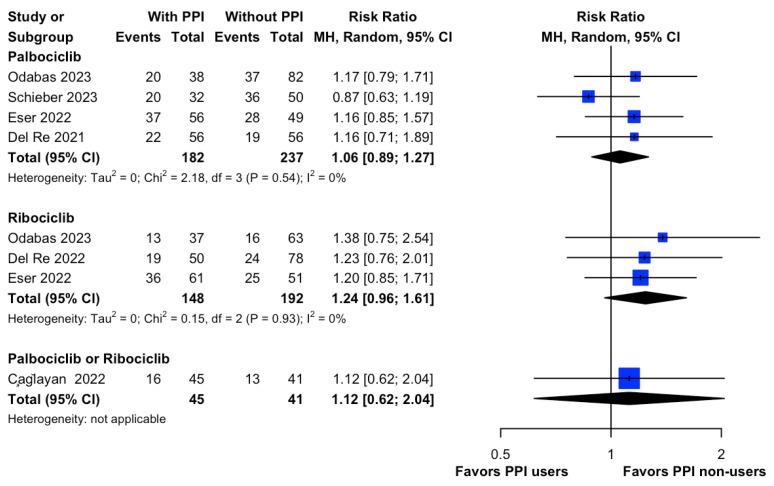
Forest plot of risk ratio of CDKI dose reduction due to drug toxicity between PPI users and non-users in HR+/HER2- metastatic breast cancer patients. The size of each blue square is proportional to the weight of the indicated study. Horizontal lines indicate the 95% CI of each study; the black diamond indicates the pooled estimate of all studies after meta-analysis with a 95% CI. TE is used to show the estimate of the treatment effect. CDKI, cyclin-dependent kinase inhibitor; PPI, proton pump inhibitor; CI, confidential interval; HR, hazard ratio; IV, inverse variance; SE, standard error; MH, Mantel–Haenszel. References: Odabas (2023) [18], Schieber (2023) [19], Eser (2022) [20], Del Re (2021) [16], Del Re (2022) [28], Çağlayan (2022) [17].

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
