# Peer review of "The Association between Proton Pump Inhibitors and the Effectiveness of CDK Inhibitors in HR+/HER- Advanced Breast Cancer Patients: A Systematic Review and Meta-Analysis"

_cancers, 2023, doi:10.3390/cancers15215133_

Round 1
Reviewer 1 Report
This is an interresting paper dealing with the negative impact of PPI users in the treatment of breast cancer HER-/HR+ patients with CDKi.
In the introduction it lacks a explication of the rational for using only postive hormone receptor patients and not also triple negative patients.In the results could you please explain better the notion of favors and it seems that the figures 3 and 4 are not well legended. What is the meaning of the color and form of the dot plots. It conducts the reader to confusion. Please clear this point.
lines 183 to 185 are not really comprehensive. Who do you call "non patient"ECOG bias not visible in table , please better define this point
Although the study tends to suggest that a possible reduction of the CDKi absorption in basis of PH variation is the reason of negative impact of PPi on survival and CDKi effectiveness. It is not clear about the bioavaibility of the two drugs since it seems that the data rather suggest that the concentrations in plasma are the same under PPI use. This point is not clear enough please try to clarify.
Line 258 vitro instead of virto, please check syntax errors through the manuscript
Reviewer 2 Report
With pleasure, I read the paper titled “Association of proton pump inhibitors and the effectiveness of CDK inhibitors in advanced breast cancer patients: A systematic review and meta-analysis”. The topic is clinically relevant to practice, and of importance to the readers of the journal CANCERS. Overall, the manuscript reads well and has good flow of ideas, up-to-date citations, and proper summary of data using tables and figures. The introduction section was detailed enough. The methods section was detailed too, however, some edits are needed for complete reporting. The research had some unavoidable limitations, all of which had been explicitly acknowledged, however, additional limitations should be also added for better transparency (see my comments below). The conclusion is line with the presented results. All in all, this manuscript is clinically significant and is very likely to be cited extensively in the future. I strongly recommend the manuscript to be accepted—however, prior to that, some changes are required as indicated below:
TITLE
· Since you only included HR+/HER- cancers, please include that in the title.
INTRODUCTION
· Moreover, it is very important to highlight the significance of your research. To my knowledge and quick database screening, this is the first-ever meta-analysis on the topic. If so, please clearly indicate that.
METHODS
· Please mention if specific filters (such as year of research, country of publication, or English language) were used during literature screening.
· Did you examine the grey literature for some additional studies?
· Did you manually screen the reference lists of recent reviews or reference lists of the included meta-analyzed studies for other potentially relevant studies that could have been missed during literature search?
· The section on how study selection was completed is missing. Please complete it. For example, did you first screen titles/abstracts and then followed by full-text screening?
· For assessment of between-study heterogeneity, have you also considered the p-value of the Cochran’s Q test (i.e., p<0.1).
· Mention the name of software used for analysis.
· The authors may need to perform leave-one-out sensitivity analysis to examine the robustness of the pooled effect sizes and check if they are impacted by single studies.
· The authors may need to provide the certainty of evidence for each outcome according to GRADE (Grading of Recommendations, Assessment, Development, and Evaluations) approach.
RESULTS
· For articles added through additional search, please provide more information.
· For Table 1, please include the country and sample size for PPI and non-PPI groups.
· For quality assessment, please summarize the data using figures for RCTs and non-RCTs.
· For data pertaining to section “3.5. CDKi dose reduction”, please refrain form using terms like “…showed little difference…” and replace with “no significant difference”.
DISCUSSION
· Please discuss the impact of PPI or no PPI on efficacy of CDKi in other cancer contexts (beyond breast cancer), if any.
· Please include a brief description of the clinical implications and future directions.
· Please acknowledge additional limitations, such as: (a) some outcomes had high heterogeneity which could be ascribed to differences in study durations, patient characteristics, and control groups, and (b) publication bias could not be assessed reliably as the number of included studies per outcome was less than the required cutoff (n=10).
Overall
· The manuscript needs some minor polishing for English language and editing.
Minor English editing is required.
Round 2
Reviewer 2 Report
The authors did a great job by attending to all comments satisfactorily. The manuscript is now scientifically sound, methodologically robust, and intellectually curious. I strongly recommend the acceptance of the manuscript in its current form.
None to minor English editing may be needed